# A Network Analysis of Emotional Intelligence in Chinese Preschool Teachers

**DOI:** 10.3390/bs14121132

**Published:** 2024-11-25

**Authors:** Sha Xie, Beiyi Su, Siman Yang, Jing Li, Hui Li

**Affiliations:** 1Faculty of Education, Shenzhen University, Shenzhen 518060, China; xiesha@szu.edu.cn (S.X.); 2210489004@email.szu.edu.cn (B.S.); 2310489013@email.szu.edu.cn (S.Y.); 2Faculty of Education, The Chinese University of Hong Kong, Hong Kong SAR, China; 3Department of Early Childhood Education, The Education University of Hong Kong, Hong Kong SAR, China; huili@eduhk.hk

**Keywords:** emotional intelligence, network analysis, early childhood education

## Abstract

Emotional intelligence significantly influences various aspects of teachers’ professional and personal lives, notably affecting preschoolers’ social skills and emotional development during formative years. This study utilizes a network analysis approach to explore the complex relationships among four components of emotional intelligence: emotional labor, emotional regulation, professional well-being, and professional identity. Participants included 2069 frontline Chinese teachers (34 males, 2035 females; M = 28.64, SD = 5.98; M years of teaching = 6.88, SD = 5.29) with no leadership roles, categorized into three stages of their careers based on years of teaching experience: novice (0–3 years; *n* = 612), advanced beginners (4–6 years; *n* = 537), and competent (7–40 years; *n* = 920). Findings revealed that joy of teaching, role value, and professional value were identified as the most critical elements within the emotional state network of early childhood education teachers. The strongest connections in teachers’ emotional networks were found between school connectedness and joy of teaching (*r* = 0.474), surface acting behavior and natural acting behavior (*r* = 0.419), and professional value and professional behavior (*r* = 0.372). Furthermore, teachers across different career stages exhibited similar characteristics and intrinsic connections among emotional state components. These findings deepen our understanding of the emotional state networks of ECE teachers, highlighting shared features and interconnected mechanisms, and suggest that enhancing teachers’ emotional intelligence through targeted professional development can improve both teacher well-being and preschoolers’ social–emotional outcomes. Policies that foster strong school connectedness and reduce emotional labor are key to promoting sustained joy in teaching, particularly for novice and advanced beginner teachers.

## 1. Introduction

Emotional intelligence (EI) has evolved from a concept primarily confined to psychology to a critical aspect of the educational sphere, particularly concerning its impact on teachers. Extensive research has highlighted EI’s pivotal role in shaping various dimensions of teachers’ professional and personal lives, including their job satisfaction [1,2], efficacy [3], and burnout [4,5]. Notably, the significance of EI is especially pronounced for preschool educators, as their EI profoundly influences the social skills and emotional development of young learners during their formative years [6,7]. The critical period of early childhood, spanning from birth to 8, is foundational for developing core competencies such as social skills and emotional regulation, which have enduring effects on future developmental trajectories [8,9]. Thus, preschool teachers, who serve as role models for positive behavior, must possess a robust repertoire of EI to effectively guide children toward healthy growth [10].

However, existing research falls short of comprehensively delineating the internal dynamics of teachers’ EI. Despite its multifaceted nature, which encompasses dimensions such as self-awareness, self-regulation, social awareness, relationship management, empathy, and motivation [11,12,13,14], there remains a notable gap in understanding how these components interconnect within the EI framework, particularly in the context of preschool educators. Questions persist regarding which factors are central to this network and whether variations exist in EI networks across different career stages. Addressing these inquiries is crucial for elucidating the fundamental elements essential for enhancing EI among preschool teachers.

Consequently, this study aims to leverage network analysis—a burgeoning statistical methodology—to uncover the core characteristics of EI. By scrutinizing the EI networks of preschool teachers, this study seeks to identify the pivotal elements within these networks and discern potential disparities across various career stages. This investigation aspires to provide empirical insights and actionable recommendations for bolstering preschool teachers’ EI, thereby contributing substantively to both empirical and theoretical advancements in EI research within educational contexts.

## 2. Literature Review

### 2.1. Conceptualization of Emotional Intelligence

EI is defined is commonly defined as the ability to effectively manage affective information, encompassing a set of emotional and social skills that influence how individuals perceive and express themselves, build and sustain social relationships, cope with challenges, and use emotional information in a purposeful and constructive manner [15,16]. It has been conceptualized as a trait [17,18] or an ability [15,19]. Within educational contexts, EI is increasingly recognized as essential for fostering positive learning environments and supporting teachers’ professional and emotional resilience [20,21]. Studies suggest that specific emotional competencies directly impact teachers’ ability to navigate classroom challenges, regulate their own emotions, and maintain professional well-being [22].

The four-branch model of EI, developed by Mayer and Salovey [13], provides a comprehensive framework for understanding how individuals perceive, facilitate, understand, and manage emotions. This model has gained substantial attention in education, especially regarding the emotional experiences of teachers. Teachers’ emotional regulation—the ability to manage and control emotional responses—can be linked to the emotional management aspect of EI, as it involves the conscious regulation of emotions to achieve desirable outcomes in the classroom [23]. Similarly, emotional labor, which refers to the effort expended by teachers to meet emotional expectations in their professional roles, aligns with the emotion facilitation and regulation components of EI, as teachers must often suppress or enhance emotions to create supportive learning environments [24]. Additionally, teachers’ professional identity, shaped by their emotional interactions in educational settings, plays a crucial role in the emotional understanding and perception branches of EI, as it influences how they interpret and respond to both their emotions and those of their students [25]. Finally, the connection between EI and professional well-being is evident, as the emotional demands of teaching require effective emotional regulation and understanding to maintain both personal and professional balance [26]. As educators strive to create positive learning environments, understanding how emotional regulation, emotional labor, professional identity, and well-being interconnect becomes critical to improving teacher performance and overall educational outcomes [27].

### 2.2. Preschool Teachers’ Emotion Regulation

Emotion Regulation (ER) denotes a cognitive approach to managing emotionally stimulating information, generally including cognitive reappraisal and expressive suppression [28,29]. Emotion regulation in teachers refers to the ability of educators to effectively manage and control their emotions in professional contexts [30]. It is positively associated with their positive effect, job satisfaction, burnout, and personal accomplishment and is essential for maintaining psychological well-being [31,32].

Early childhood educators constantly employ cognitive reappraisal strategies for emotion regulation along with their perceptions of positive relationships [33]. Cross-cultural research studies have also identified variations in teacher emotion regulation across nationalities and individual educators, with the correlation between mental health and emotion regulation being stronger among Korean teachers than among American counterparts [34]. It was found that there was a high level of negative correlation between preschool teachers’ emotion regulation skills and occupational anxiety [35]. Furthermore, the emotion regulation of early childhood educators mediates the relationship between teachers’ emotional distress and work engagement [36]. The use of emotion regulation strategies by teachers to alleviate children’s emotional states is linked with increased levels of progressive autonomy promotion and occupational well-being [37].

Moreover, children are notably influenced by the emotion regulation abilities of their early childhood educators. Teachers’ capacity for emotion regulation is related to the occurrence of maladaptive behaviors in preschool students [38], and they engage in challenging social interactions as part of their efforts in emotion regulation [33]. Therefore, it is crucial to deliver effective social–emotional classroom practice programs in teacher education, given that they are crucial for enhancing teachers’ emotion regulation strategies [39]. Exploring early childhood teachers’ emotion regulation holds significant implications for early childhood education (ECE) practice.

### 2.3. Preschool Teachers’ Emotional Labor

The concept of emotional labor, originally introduced by Hochschild in 1983 within the workplace, pertains to the management and expression of emotions through individuals’ behaviors, either superficially or profoundly [40]. Grandey proposed that emotional labor strategies include two kinds of strategy: surface behavior and deep behavior [41]. Since its inception, emotional labor has been extended to the realm of education, particularly within early childhood teaching, where it is characterized by its prolonged duration, intensity, and diverse emotional interactions [42]. Yin et al. [32] summarized three categories and seven strategies for Chinese teachers to regulate emotions in the classroom: (1) surface acting (pretending and restraining); (2) deep acting (refocusing, reframing, and separating); and (3) genuinely expressing (releasing and outpouring). Notably, studies indicate a prevalence of profound emotional behaviors among early childhood educators, surpassing the frequency of superficial displays [43]. The emotional labor exerted by these educators profoundly influences both the classroom environment and the social emotional development of children [44].

A multitude of factors intersect with emotional labor, including teachers’ experience levels, emotional exhaustion, and psychological capital [45]. Additionally, research suggests that teachers’ approaches to emotional labor vary across different career stages [46]. Given these complexities, targeted training initiatives aimed at enhancing emotional labor skills are imperative for early childhood teachers to bolster their teaching effectiveness [2]. Despite efforts in this direction, early childhood educators continue to encounter unique challenges associated with emotional labor when compared to their counterparts at different career stages [47]. Moreover, cross-cultural studies reveal disparities in the levels of emotional labor stress experienced by educators, underscoring the importance of contextual factors in shaping emotional labor experiences [48].

### 2.4. Preschool Teachers’ Professional Identity

Professional identity encompasses a self-concept built upon attributes, beliefs, values, motivations [49], and experiences reflecting an individual’s inclination to remain in their profession and their level of satisfaction [50]. It is a dynamically evolving construct [51,52], influenced by various factors, such as personal attributes like educational background [53], teaching experience [54,55], and age, resulting in differing levels of professional identity [56].

Furthermore, encouragement, support, and understanding from friends and family play pivotal roles in fostering preschool teachers’ professional identity, aiding in alleviating physical and mental pressures, and enabling them to engage more effectively in their work [57,58]. In turn, the professional identity of preschool teachers can significantly impact their personal well-being, job satisfaction, burnout, and self-efficacy [59,60].

Recent research has also underscored the close interplay between preschool teachers’ professional identity and emotions [61,62]. On one hand, teachers’ emotions impact their professional identity, with positive emotions bolstering it [62,63,64], while negative emotions can diminish it [63,65] or prompt a reevaluation of their professional identity [64]. On the other hand, professional identity can also influence teachers’ emotions, with strong profession identification leading to greater happiness derived from work, increasing liking for their occupation, and heightening commitment to realize the profession’s vision, thereby shaping their emotional experiences [66,67]. However, few studies have analyzed professional identity as a relevant factor of EI, indicating the need for further exploration in the current study.

### 2.5. Preschool Teachers’ Professional Well-Being

Teachers ‘professional well-being is interpreted as a positive evaluation of all aspects of their work, including emotion, motivation, behavioral, cognitive and physical and mental dimensions [26,68], and is one of the important factors determining teachers’ professional growth, success, motivation and professional activities [26]. With the increasing attention of society to ECE, the professional well-being of preschool teachers has gradually become the focus of research. Numerous studies have shown that professional well-being is closely related to job performance [69], ability to provide quality education and supportive classroom atmosphere [70,71,72], and willingness to stay in ECE [73].

However, there are relatively few studies on the professional well-being of preschool teachers, and there is a lack of in-depth and systematic discussion. In the existing research, the researchers have analyzed and discussed the professional well-being of preschool teachers from several perspectives. On one hand, elements of professional well-being about preschool teachers are considered as a broad concept involving both positive and negative indicators, and key factors include job satisfaction, job stress, work engagement, and job burnout [74,75,76,77,78]. On the other hand, relevant studies also explore the factors that affect preschool teachers’ professional well-being, which are mainly divided into personal factors and environmental factors. In terms of personal factors, self-efficacy affects the development of preschool teachers’ professional well-being [79], and for the environmental factors, mainly with working conditions [74,79,80], working pressure [72,81], and interpersonal relationships [82,83]. Although the professional well-being of preschool teachers warrants further exploration, existing studies underscore its significant influence on the quality of early childhood education and the well-being of young children [74,84].

### 2.6. Preschool Teachers’ Emotional Intelligence at Different Career Stages

Education is inherently enriched by human interactions and emotional exchanges, making teaching a profoundly emotionally engaging activity [85]. Numerous studies have investigated the emotional behaviors displayed by preschool teachers at different career stages. For instance, Terzioğlu and Tuğçe Esra found that teachers with varying tenures (ranging from 8.5–11 years, 11.01–16 years, to 16.1–29 years) exhibited distinct abilities in nurturing children’s emotional capabilities [86]. In another study, 394 Chinese preschool teachers were categorized into five distinct groups based on their teaching experience and found that teachers of different teaching ages exhibited different degrees of burnout, and that EI mediated the relationship between teachers’ positive thinking and burnout [87].

Despite the growing attention on teachers’ EI in current research, there is a notable lack of studies exploring the EI of preschool teachers across different years of teaching experiences, which varied significantly. Therefore, this thesis aims to bridge this gap by categorizing the study population into distinct career stages for further investigation.

### 2.7. Network Analysis of Emotional Intelligence

The method of network analysis aims to represent the characteristics and information of a system in the form of a network, consisting of “nodes” and “edges”. Traditionally, nodes represent entities (such as neurons, stations, or individuals), while edges denote connections between these entities (such as synapses, routes, or interpersonal relationships) [88,89]. With the advent of data analysis approaches, the evolving characteristics of nodes and edges reflect the dynamics of the network. In contrast to traditional models, network analysis based on observed variables (referred to as network analysis hereafter) assigns observed variables like attitudes, feelings, and behaviors to nodes, with edges representing connections between these variables [90]. This approach provides quantitative indicators to measure the importance of nodes within the network, facilitating the identification of central nodes that can activate others and influence the entire network [91,92,93].

However, there are limited studies utilizing network analysis to explore EI in teachers. Ramos-Vera et al. [94] employed network analysis to examine the EI and burnout dimensions among Peruvian teachers, identifying significant connections between emotional exhaustion, depersonalization, and low self-actualization, as well as between understanding others’ emotions and utilizing emotions. This study underscored the significance of evaluating others’ emotions and utilizing emotional components in strengthening the dynamic link between EI and reduced burnout, particularly in mitigating emotional exhaustion. Such investigations into teachers’ EI are pivotal for enhancing intervention and prevention strategies against teacher burnout.

Apart from the study of Ramos-Vera et al. [94], existing studies have explored the network structure of EI in other populations. Stochl et al. [95] explored the network structure of the Warwick–Edinburgh Mental Health Scale across four British cohorts, highlighting the centrality of items related to positive self-perception and emotion. Fisher et al. [96] utilized network analysis to investigate mood and anxiety dynamics in patients with generalized anxiety disorder or major depressive disorder, identifying core symptoms associated with positive and negative emotions.

While these studies have advanced our understanding of EI and offered insights into prevention and intervention strategies, several research gaps persist. Notably, there is currently no systematic review of preschool teachers’ EI networks, despite the increasing attention to teachers’ EI. Furthermore, research in the field of ECE, particularly in non-Western countries like China, remains scarce, necessitating further investigation given the distinct differences in ECE systems and policies. Moreover, existing research often focuses on a subset of EI elements, failing to fully capture the complexity of EI networks. Future studies incorporating a broader array of EI elements in network analysis are warranted to address these gaps and facilitate comprehensive advancements in our understanding of EI.

### 2.8. The Current Study

Kindergartens in China serve as institutions offering care and education for children aged 3 to 6 years old, typically spanning a three-year program divided into junior, middle, and senior classes [97], which is the focus of the current study. With the steady growth in the number of kindergartens in China, there is an escalating demand for highly skilled teachers. The release of the *Guidelines for the Evaluation of Kindergarten Care and Education Quality* (幼儿园保育教育质量评估指标) in 2022 has underscored the need for teachers to enhance their effectiveness in education and pursue ongoing professional development within the teaching workforce [98]. As of December 2023, the release of the *Measures for Supervision and Evaluation of Kindergartens* (幼儿园督导评估办法) has further underscored that supervision should primarily emphasize assessing teachers’ abilities to cultivate a nurturing emotional atmosphere and maintain a positive and optimistic emotional disposition [99].

By 2023, China’s gross preschool enrollment rate has reached 91.1%, up from 56.6% in 2010. This achievement is notable, considering the larger class sizes in Chinese kindergartens compared to other countries, as well as the parental expectations for high-achievers faced by teachers [100]. Against this backdrop, it becomes imperative to delve deeply into the EI of early childhood teachers. Thus, this study aims to employ a network analysis approach to investigate the intricate relationships among these four factors of EI: emotional labor, emotional regulation, professional well-being, and professional identity. Specifically, this study addressed the following two research questions:What are the core elements of preschool teachers’ EI network?Does the network vary or exhibit similarities among different career stages?

## 3. Materials and Methods

### 3.1. Participants

Participants in this study were recruited from 123 kindergartens located in Shenzhen, China (see Table 1). Among these, 83 were classified as public schools, as they received government subsidies. A total of 2345 responses were collected through paper questionnaires. However, 276 responses were either incomplete or duplicated, resulting in 2069 valid samples for analysis (34 males, 2035 females; mean age = 28.64, SD = 5.98; mean working year = 6.88, SD = 5.29). The proportion of male teachers in the study is representative of that in the country, being less than 1%. All participants were front-line teachers without holding leadership positions, distributed across teaching levels K1 (*n* = 691), K2 (*n* = 674), K3 (*n* = 694) and 10 teachers teaching mixed-age classes. Following Booth et al.’s [101] model, participants were categorized into five career stages: novice (0–3 years; *n* = 612), advanced beginners (4–6 years; *n* = 537), competent (7–18 years; *n* = 839), proficient (19–30 years; *n* = 77), and expert (31–40 years; *n* = 4). Due to small sample sizes in the proficient and expert stages, they were merged into the competent stage to ensure sufficient statistical power. Consequently, participants were classified into three career stages for this study.

### 3.2. Procedure

The human research ethics committee at the first author’s university granted permission to review and approve the study. A total of 123 kindergartens were randomly selected from the five administrative districts (*Luohu*, *Longgang*, *Futian*, *Nanshan*, *Longhua*) of Shenzhen. Following the principal’s consent, all teachers in each kindergarten were invited to participate in the study. Upon obtaining their consent, teachers were asked to complete the questionnaires, which typically took around 10 min on average.

### 3.3. Measures

#### 3.3.1. Professional Well-Being

This study utilized the Teacher Subjective Well-being Questionnaire (TSWQ; [102]) to assess the professional well-being of individual teachers, which has been used in the Chinese context [103]. The TSWQ comprises three subscales: school connectedness (SC; 4 items; e.g., “I can really be myself at this school”), joy of teaching (JT; 4 items; e.g., “I am very interested in the things we are doing at this school”), and teaching efficacy (TE; 4 items; e.g., “I am good at helping students learn new things”). Responses were rated on a 4-point Likert scale ranging from 1 (almost never) to 4 (frequently), with higher scores indicating higher levels of teacher well-being. The validity and reliability of the scale in this study was satisfactory: CFI = 0.942, TLI = 0.925, SRMR = 0.039, RMSEA = 0.057, and with αTotal = 0.871, Asc = 0.768, αJT = 0.763, αTE = 0.768.

#### 3.3.2. Emotional Labor

To gauge teachers’ emotional labor, this study employed the Emotional Labor Scale [104], which consists of 13 items encompassing surface acting behavior (SB; 6 items; e.g., “I show feelings to students or their parents that are different from what I feel inside”), expression of naturally felt emotions (NB; 4 items; e.g., “the emotions I show students or their parents match what I spontaneously feel”), and deep acting behavior (DB; 3 items; e.g., “I make an effort to actually feel the emotions that I need to display towards students or their parents”). Participants rated their responses on a 5-point Likert scale ranging from 1 (strongly disagree) to 5 (strongly agree), with mean scores computed for each subscale. The validity and reliability of the scale were confirmed in the current study: CFI = 0.938, TLI = 0.921, SRMR = 0.058, RMSEA = 0.072, and with α*_Total_* = 0.794, α*_SB_* = 0.857, α*_NB_* = 0.817, α*_DB_* = 0.866.

#### 3.3.3. Emotion Regulation

The current study assessed teachers’ emotion regulation abilities using the Emotion Regulation Scale [105], which includes reappraisal (RE; 5 items; e.g., “When I’m faced with a stressful situation, I make myself think about it in a way that helps me calm down”) and suppression (SU; 4 items; e.g., “When I am feeling negative emotions, I make sure not to express them”). Participants rated their responses on a 5-point Likert scale ranging from 1 (never/strongly disagree) to 5 (very often/strongly agree), with higher scores indicating better emotion regulation. The scales demonstrated good validity: CFI = 0.963, TLI = 0.948, SRMR = 0.031, RMSEA = 0.05, and satisfactory internal consistency: α*_Total_* = 0.69, *α_RE_*= 0.755, *α_SU_* = 0.799.

#### 3.3.4. Professional Identity

Teachers’ professional identity was measured using the Professional Identity Scale [106], consisting of 18 items across four dimensions: role values (RV; 6 items; e.g., “I will realize the value of my life as a teacher”), professional behavioral tendencies (RB; 5 items; “I believe that the teaching is important for the individual development of human beings”), professional values (PV; 4 items; e.g., “I adhere to professional standards to maintain order in the school”), and professional belongingness (PB; 3 items; e.g., “I feel offended when the teaching profession is criticized unfairly”). Responses were rated on a five-point Likert scale from 1 (very non-conforming) to 5 (very conforming), with higher scores indicating a stronger professional identity. Mean scores were computed for each dimension. The scale demonstrated good validity and reliability: CFI = 0.918, TLI = 0.901, SRMR = 0.05, RMSEA = 0.054, and with αTotal = 0.886, αRV = 0.842, αRB = 0.719, αPV = 0.769, αPB = 0.691.

### 3.4. Data Analysis

Utilizing a network analysis approach, the study aimed to reveal the interconnectedness among various components of preschool teachers’ emotion status, including well-being, emotion labor, emotion regulation and professional identity. Initial analyses involved descriptive assessments, bivariate correlations, and independent *t*-tests, conducted using SPSS 28.0. The network itself was constructed using R 4.1.2 [107], employing the graphical least absolute shrinkage and selection operator (GLASSO) estimation algorithm with Extended Bayesian Information Criterion (EBIC glasso) in the Gaussian Graphical Model [108,109]. The network analysis involved four main steps: network estimation, centrality assessment, stability evaluation, and network comparison. The first three steps primarily focused on addressing the primary research question, while the final step addressed the secondary question. Data and code for this study can be obtained by contacting the first author.

#### 3.4.1. Network Estimation

Following the network estimation, the bootnet package (version 1.5) [107] was utilized to estimate the entire emotion status network. Subsequently, the qgraph package (version 1.9.2) [110] was employed to visualize the network. Each node within the network represents a specific emotion status element, while the connections between nodes are depicted as edges. The graphical representation of the network provides insights into the centrality and proximity of nodes to each other. Shorter edges between nodes indicate stronger relationships, with blue lines representing positive associations and red lines denoting negative associations.

#### 3.4.2. Network Centrality

Centralities in network analysis encompass node expected influence (EI), strength, closeness, and betweenness [108,111,112]. Due to the instability of closeness and betweenness in psychological networks, node EI and strength are primarily utilized [111]. Initially, to assess how well a node is directly connected to other nodes in the network, the centrality measure EI was explored using the qgraph package [110]. This metric includes the absolute sum of all edges, considering the presence of negative associations. Subsequently, to identify influential nodes, degree centrality as the sum of all absolute incoming edge strengths of a particular node was computed. Thirdly, closeness was computed, indicating that a node with stronger closeness may influence all other nodes more rapidly. Lastly, betweenness was computed, which denotes the frequency with which a node lies on the shortest path between two other nodes [113].

Furthermore, node predictability was computed using the R package mgm [114]. This metric evaluates the interconnectedness of a particular node with the rest of the network by measuring its shared variance with surrounding nodes. Unlike centrality, which represents a relative measure of node influence, predictability offers an absolute metric. Essentially, predictability quantifies the variance explained in a given node by all other nodes in the network, whereas strength gauges interconnectedness relative to other nodes in the network.

#### 3.4.3. Network Stability

In gauging the network’s stability, the bootnet package was utilized to assess the centrality metrics’ stability through computation of the correlation stability coefficient (CS coefficient). Typically, a CS coefficient of 0.70 signals the highest acceptable sample reduction, while coefficients surpassing 0.50 are generally deemed adequate, adhering to a minimum threshold of 0.25 [108].

#### 3.4.4. Network Comparison

For comparing the global connections across different career stages within the networks, the R package NetworkComparisonTest [115] was utilized. This comprehensive test evaluates disparities between networks based on various invariance measures, encompassing network structure, global strength, network invariance, centrality indices and predictability. Specifically, the global strength invariance test determined whether the overall connectivity remained uniform across networks, while the network structure invariance test assessed the similarity of structure between two networks. Furthermore, the centrality invariance test explored variations in specific centralities. Our network comparison test involved three distinct subgroups: the novice stage, the advanced beginner stage, and the competent stage, with comparisons made among each subgroup.

### 3.5. Missing Data

Before initiating the network analysis, missing data were addressed through random forest estimation, a technique facilitated by the missForest R package [116].

### 3.6. Covariates

Critical demographic variables, including teachers’ teaching grade, gender, position, education attainment, and whether they took lunch breaks or engaged in overtime work, were considered as covariates in the network analysis.

## 4. Results

### 4.1. Descriptive Results

The means and standard deviations, kurtosis, and skewness of the study’s variables, as well as the correlations between all study variables, appear in Table 2. Table 3 compares the differences of variables in different career stages of teachers, e.g., SB, SU, and PV do not have career stage variability, while TE and RV have significant stage differences and, the higher the career stage of a teacher, the higher his/her values in TE and RV.

### 4.2. Core Elements of Teachers’ Emotional Intelligence Network

#### 4.2.1. Regularized Partial Correlation Network

The regularized partial correlation network is depicted in Figure 1, comprising a total of 41 edges (selected lambda = 0.009). The strongest connections observed in the network were between SC and JT (*r* = 0.474), SB and NB (*r* = 0.419), and PV and RB (*r* = 0.372).

#### 4.2.2. Centrality

Figure 2 depicts the centrality indices, as visualized in Table 4. Notably, RV (1.12), JT (1.08), SB (1.04), and PV (0.97) exhibited the highest strength centrality values, indicating robust interconnectedness, while SU (0.14) and PB (0.44) had the lowest.

Expected influence (XI) was the primary focus of this study, as nodes with high expected influence might play crucial roles in the activation and maintenance of an interactive network [117]. In terms of EI centrality, JT (0.94), PV (0.93), RB (0.91), and RV (0.86) demonstrated the strongest associations with other elements, significantly higher than all others in the network in terms of strength values.

Predictability, measured by squared multiple correlation coefficients, ranged from 0.613 to 0.00. PV exhibited the highest node predictability, sharing 61% of its variance with surrounding nodes, followed by TE (54%) and NB (49%). Conversely, RE and SU did not have node predictability.

In summary, JT, RV, and PV emerged as the most crucial elements in the emotion status network among preschool teachers. These elements consistently displayed the highest values across centrality indices, indicating their pivotal roles within the network. Conversely, RE and SU consistently exhibited the lowest values.

#### 4.2.3. Results for Network Stability

Several studies have verified that node strength remains the most stable metric, whereas closeness and betweenness diminish notably as the sample size decreases [107]. The findings reveal that node strength boasts a CS values of 0.673, which exceeds 0.5, indicating remarkable stability.

### 4.3. Emotional Intelligence Network Differences Across Career Stage

The emotion status network among teachers at various career stages was compared and visualized in Figure 3, Figure 4 and Figure 5. Consistent with the findings from the entire sample, three pairs of edges consistently exhibited strong associations: JT and SC (0.447; 0.482; 0.462), PV and RB (*r* = 0.335; 0.414; 0.363) as well as SB and NB (*r* = 0.302; 0.389; 0.473).

The node centrality indices for each career stage are illustrated in Figure 6. JT and PV consistently exhibited the highest XI, strength, and predictability values across the subgroups. Centrality indices were directly compared across stages using difference tests, as summarized in Table 5 and Table 6. Strength values for the novice, advanced beginner, and competent stages were 4.570, 4.435, and 4.922, respectively. The global strength invariance tests revealed no significant differences among subgroups. However, network structure invariance tests identified significant differences between C1 and C2, and C1 and C3, while no significant differences were found between C2 and C3.

Further network comparison tests indicated no significant differences in centrality indices across subgroups. Corrected *p* values were largely insignificant, suggesting equivalence in XI, strength, and predictability for each EI element across stages. These findings indicate similar features and internal mechanisms of interrelations between emotion intelligence elements among preschool teachers at different career stages.

## 5. Discussion

In this study, the EI networks of teachers at different career stages were analyzed and compared. By utilizing the network analysis method, it was found that joy of teaching, role value, and professional value play important roles in preschool teachers’ EI networks and that teachers’ EI networks at various stages have similar characteristics and intrinsically linked mechanisms. This provides a deeper understanding of the structure of teachers’ emotions intelligence and may provide a basis for subsequent research and teacher development strategies.

### 5.1. Characteristics of Teachers’ Emotional Intelligence Network

The current study confirms that the EI of kindergarten teachers is a multidimensional construct composed of interrelated elements, including emotion regulation, emotional labor, professional identity, and professional well-being, all intertwined in a complex EI network [11,16,118]. This finding aligns with previous studies employing factor analysis [119,120,121] but extends the understanding of the intricate patterns of interrelationships among these dimensions of EI. By utilizing network analysis, the present study investigates how various factors of EI interrelate while controlling for the influence of other variables, thereby offering a more nuanced understanding of the construct of teachers’ EI.

Among the twelve elements that constitute EI, joy of teaching, role value, and professional value emerge as the three most critical factors in this network. First, joy of teaching serves as a crucial component within the EI framework, reflecting the positive emotions and satisfaction preschool teachers experience in their pedagogical practice [122,123]. This sense of enjoyment enhances teachers’ intrinsic motivation, fostering greater engagement and enthusiasm in their instructional activities. For preschool educators, finding joy in teaching is essential for promoting higher levels of professional well-being and a positive emotional state, consistent with previous research findings [124]. Second, professional value represents preschool teachers’ acknowledgment and respect for their own professional worth [125]. When educators perceive their profession as carrying significant social value and offering ample opportunities for personal development, they experience greater satisfaction and pride. This positive emotional state not only bolsters EI but also helps teachers maintain a resilient professional mindset in the face of burnout or external challenges, reducing the likelihood of career turnover—a critical manifestation of EI [55,126]. Finally, role value underscores the importance of understanding the significance that preschool teachers embody within their educational roles. A clear grasp of role value cultivates self-awareness and self-efficacy, allowing teachers to recognize their vital contributions to their students’ developmental process. As a result, they are more likely to demonstrate elevated levels of responsibility and emotional management skills [127].

The prominence of these elements within the network analysis of kindergarten teachers’ EI highlights the unique demands of their profession. Young children in the preschool stage are undergoing critical emotional development, marked by significant mood swings that necessitate teachers’ guidance and comfort with patience and compassion [128]. Given the emotional demands inherent in this field, deriving joy from teaching is crucial for sustaining educators’ long-term commitment [129]. When teachers find joy in their work, they are more likely to perceive value in their contributions, leading to enhanced professional satisfaction and overall happiness, which in turn fosters the development of EI [124].

Moreover, a significant shift occurred in China’s public policy regarding early childhood education (ECE) in the 1990s, transferring responsibility for funding and oversight from the government to the private sector and non-governmental organizations [130]. This transformation has positioned ECE within the market economy, reframing teachers as service providers rather than respected professionals [43]. Since 2010, a series of educational reform policies have been implemented to address the long-standing issues of accessibility, affordability, and accountability in ECE, aiming to provide high-quality and universal services to all families [130,131,132]. Despite these initiatives, recent studies indicate a concerning turnover rate among kindergarten teachers in China [133]. In this context, the recognition of professional value is particularly vital, as it enables educators to appreciate the significance of their work, fostering a sense of belonging that can help them maintain emotional stability and a positive outlook in the face of challenges [87].

Finally, the multifaceted nature of the teacher–child relationship significantly impacts young children’s development [134]. Preschool teachers assume various roles—supporters, collaborators, mentors, role models, and demonstrators—that demand a high degree of EI. This capacity allows teachers to respond flexibly to the emotional challenges that arise in diverse situations [127,129,135]. By understanding the intricate dynamics of their EI network, educators can enhance their effectiveness in nurturing the emotional and social development of their students.

### 5.2. The Fabric of Emotion: Correlations in ECE Teachers’ Emotional Intelligence

Early Childhood Education (ECE) teachers’ emotions represent a multidimensional structure comprised of interrelated factors, forming a complex emotional network. Among the twelve factors relevant to teachers’ emotions, three pairs of elements demonstrated particularly strong correlations, shedding light on the dynamics of EI in this context.

Firstly, the study revealed a significant correlation between teachers’ sense of school connectedness and their joy of teaching. This suggests that the degree of belonging teachers perceive within the preschool environment is intricately linked to the enjoyment they derive from their teaching. Social Identity Theory posits that individuals tend to evaluate positively and feel emotional attachment to the groups to which they belong, which can translate into a favorable attitude and increased enjoyment of work [136]. The care and support provided by preschool leaders, along with understanding and assistance from colleagues, amplify teachers’ connections to their institutions. This connection fosters a sense of spiritual comfort, allowing teachers to maintain positive emotions and experience the joy and fulfillment inherent in their teaching roles [137,138]. Therefore, enhancing teachers’ well-being is crucial for promoting their sense of belonging and joy in the teaching profession [76].

Secondly, the current study identified a strong correlation between surface acting and natural acting behaviors, likely reflecting the influences of China’s collectivistic culture. In this cultural context, preschool teachers often prioritize the collective interests of the school by suppressing negative emotions to uphold positive relationships with students, parents, and colleagues [139]. Consequently, teachers may conceal feelings of frustration or disappointment, continuing to teach with enthusiasm despite internal conflicts. For instance, when interacting with parents or addressing misbehavior in students, teachers might mask feelings of disrespect or frustration to avoid confrontation and maintain classroom harmony [140,141]. The Job Demands–Resources (JD–R) model illuminates this behavior further [142,143]; while high job demands can lead to emotional exhaustion, the presence of abundant job resources—such as support from leaders and colleagues—can alleviate stress and encourage positive emotional expressions, whether surface-level or authentic, that align with organizational expectations [141,144]. Thus, preschool teachers may engage in surface acting to fulfill job demands, while supportive resources help mitigate the associated emotional strain [145].

Finally, a robust correlation was observed between professional behavior and professional values, indicating that preschool teachers’ values can significantly motivate and be reflected in their actions [146]. Van den Berg highlights that professional identity emerges from the interplay between teachers’ individual experiences and their broader social, cultural, and institutional environments [147]. In China, for example, teachers are culturally and ethically expected to demonstrate care for their students. Guided by these professional norms, ECE teachers frequently engage with young children, exhibiting a high degree of emotional investment in their work [148]. This commitment strengthens their understanding of their roles, solidifies their dedication to professional values, and nurtures a strong sense of professional mission that is clearly reflected in their teaching practices [149].

Overall, these findings underscore the importance of EI among ECE teachers, revealing how critical elements are intricately linked within a network that influences their professional experiences and interactions. Understanding these correlations provides valuable insights into how educators can enhance their EI and improve their teaching efficacy.

### 5.3. Teachers’ Emotional Intelligence Network Varies Across Career Stages

The current study found that, while significant differences in clustering patterns were observed across various career stages—most notably between novice and competent teachers and between novices and advanced beginners—the importance of central nodes remained relatively consistent across these subgroups.

Firstly, distinct differences were noted in the EI networks of novice versus experienced preschool teachers. The development of EI is a dynamic process shaped by both experience and training [15,150,151]. As teachers gain experience, they typically become more adept at integrating into their school’s culture, fostering stronger teacher–child relationships, and deriving increased satisfaction from their work [152,153,154]. Advanced and competent teachers navigate a wider array of teaching situations and continuously refine their instructional strategies through critical reflection [155]. This ongoing reflective practice enhances their emotional management and teaching efficacy [155]. Generally, these more experienced teachers benefit from extensive professional development opportunities, which deepen their expertise and contribute to improvements in self-efficacy, job satisfaction, and overall professional performance [156,157]. It is noteworthy that advanced and competent teachers are usually more proficient in deep acting behaviors, whereas novice teachers tend to exhibit more natural acting behaviors. This disparity may be attributable to novice teachers’ ongoing adaptation to their roles and environments, which significantly influences their emotional responses and behaviors [158]. Consequently, it is essential for novice teachers to continue their professional development to enhance both their teaching skills and EI.

Secondly, the centrality index of EI components across all three career stages was comparable, with no significant differences between subgroups. This finding suggests that, regardless of career stage, core elements of teachers’ EI—such as joy in teaching, role value, and professional worth—maintain a similar central position within the EI network. This underscores the consistent importance of these components throughout a preschool teacher’s career.

The stability of these EI components highlights the continuity of preschool teachers’ professional development. It implies that, despite experiencing various career changes and challenges, the essential professional sentiments and values among teachers remain relatively stable. This stability is critical for fostering job satisfaction, enhancing work performance, and securing long-term professional growth. According to Bar-On’s EI model (Bar-On, 2006), mood components, such as optimism and happiness, are vital for managing environmental demands and pressures [11]. Teachers’ ability to maintain a positive attitude and derive satisfaction from their work plays a significant role in mitigating stress, which may explain why feelings of joy and happiness in teaching remain consistent across different career stages [93,159].

Cultural factors may also contribute to this consistency. In China, the caring ideology rooted in traditional cultural values surrounding childcare influences teachers’ professional outlooks [160]. This cultural context emphasizes the importance of emotional competencies and self-actualization, aligning with Maslow’s hierarchy of needs [161]. As such, regardless of career stage, preschool teachers’ self-regard and self-actualization are crucial for effectively managing emotional challenges and reinforcing their identification with their roles and professional values.

Overall, these insights emphasize the importance of nurturing EI throughout the teaching career, ensuring that preschool educators can thrive in their roles and effectively support the development of their young learners.

## 6. Limitations, Implications, and Conclusions

Before delving into our findings, itis crucial to acknowledge several limitations of our study. Firstly, it was cross-sectional, limiting our ability to discern the direction of associations among teachers’ mood state subgroups or how these dimensions interact over time [162]. Prospective and longitudinal studies are necessary to uncover differential associations between subgroups. Secondly, our data were sourced from 120 schools, potentially introducing nesting effects, though our analyses didn’t account for this multilevel structure due to technological constraints. While contemporary network analysis techniques can handle time-series data, the absence of repeated observations in our dataset limits the suitability of the current approach. Future research could refine network analysis methods to better accommodate nested data structures. Thirdly, the network structure’s shape hinges on specific node variables [92], and the exclusion of crucial nodes, like teachers’ emotional knowledge and beliefs in our study, may compromise inferences about centrality. Subsequent studies should incorporate more comprehensive indicators of teachers’ emotional development to provide a deeper understanding of key features within teachers’ emotional states. Lastly, network analyses may be influenced by demographic and cultural disparities. The underrepresented male teachers in the sample might show different network patterns in comparison to the female teachers, which should be explored in the future research from the perspective of male educators in preschool settings. Furthermore, endeavors to gather data from diverse cultural contexts should entail a thorough assessment of teachers’ emotional states.

The implications of the current study warrant mention. Theoretically, the current study adds depth to existing theories of EI by highlighting the centrality of joy of teaching, role value, and professional value within the EI network of preschool teachers. This finding enriches the understanding of how these elements interplay and contribute to the overall emotional experience of educators, particularly within the Chinese cultural context. Practically, the current study can inform the design of professional development programs aimed at enhancing teachers’ EI, by reinforcing joy in teaching and acknowledging the value of teachers’ roles. Given the study’s focus on Chinese preschool teachers, the findings underscore the importance of cultural context in EI research. Practical applications should consider cultural values and expectations, which can influence teachers’ emotional experiences and professional identity.

In conclusion, through the application of network analysis methodology, the research aims to identify the central elements within the framework of teachers’ emotional states, considering multiple factors concurrently. Our findings underscore that joy of teaching, role value, and professional value emerged as crucial components of the emotional state network within the Chinese context. Despite variations in the relative importance of different dimensions of teachers’ emotional states across various career stages, the significance of these three elements remains consistent throughout. These insights make a unique contribution to theories focusing on EI and provide valuable guidance for developing intervention strategies to foster the emotional growth of teachers.

## Figures and Tables

**Figure 1 behavsci-14-01132-f001:**
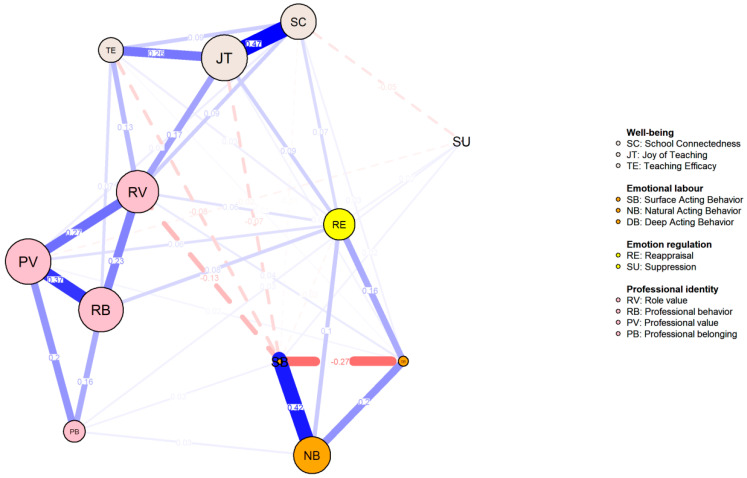
Network for the Whole Sample.

**Figure 2 behavsci-14-01132-f002:**
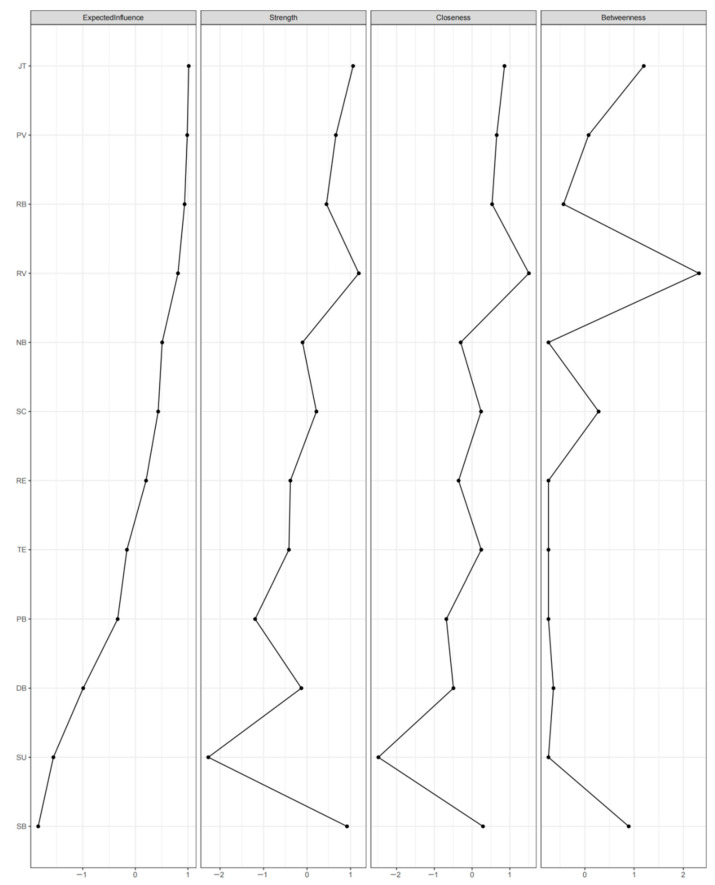
Node Centrality for the Whole Sample. Note. JT = Joy of Teaching; PV = Professional value; RB = Professional behavior; RV = Role value; SC =School connectedness; NB = Natural acting behavior; RE = Reappraisal; TE = Teaching efficacy; PB = Professional belonging; DB = Deep acting behavior; SU = Suppression; SB = Surface acting behavior.

**Figure 3 behavsci-14-01132-f003:**
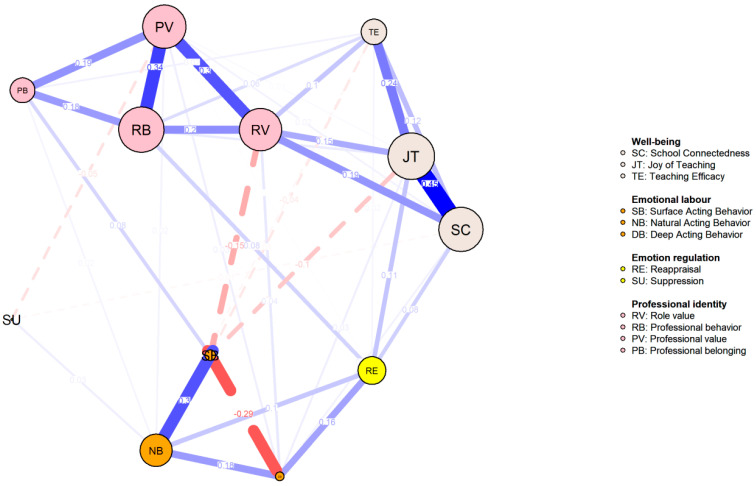
Network for the Novice Stage.

**Figure 4 behavsci-14-01132-f004:**
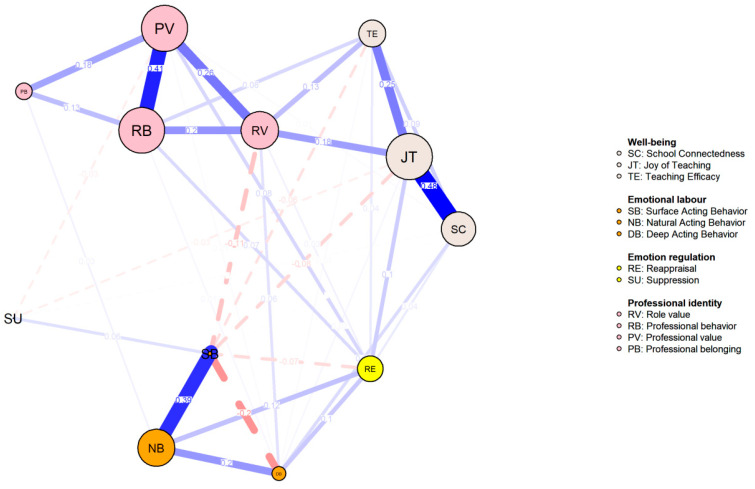
Network for the Advanced Beginners Stage.

**Figure 5 behavsci-14-01132-f005:**
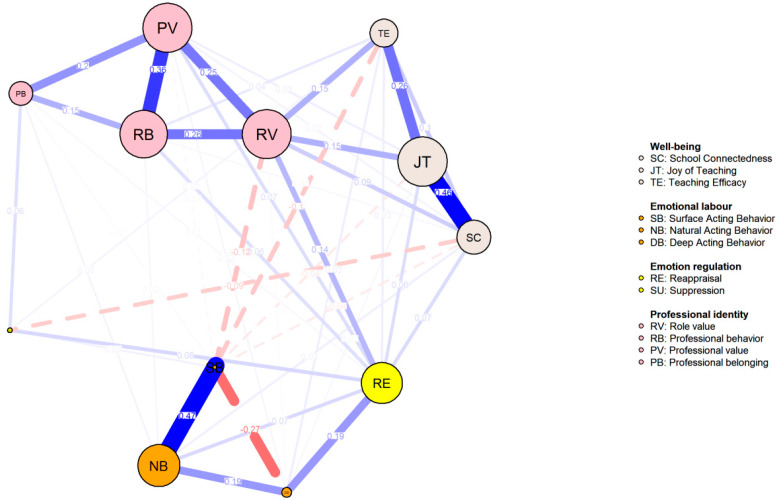
Network for the Competent Stage.

**Figure 6 behavsci-14-01132-f006:**
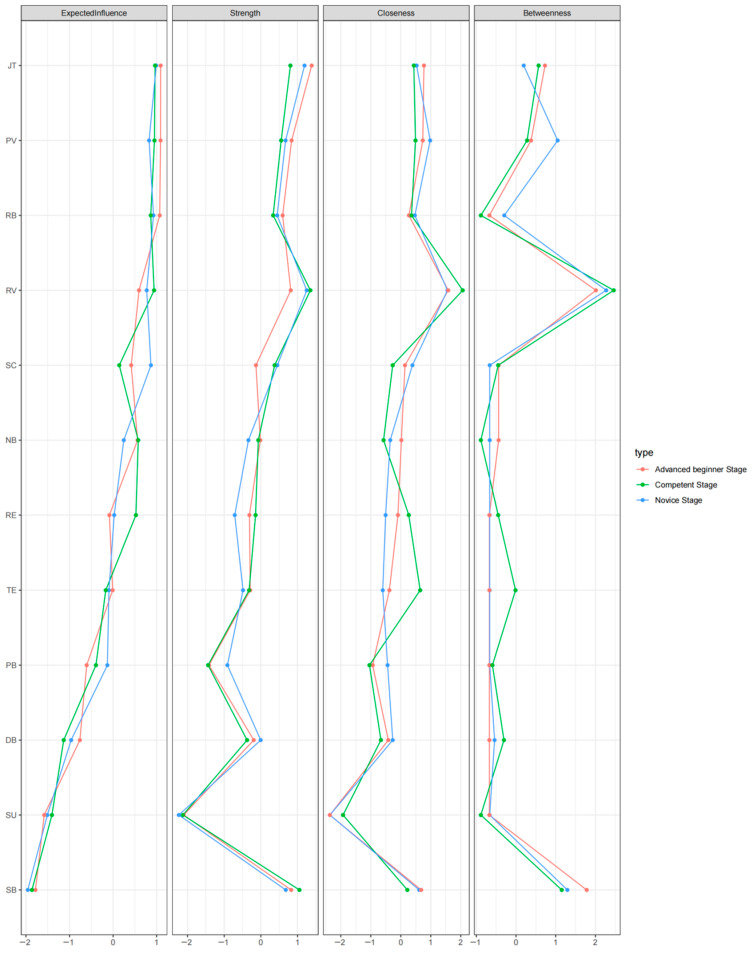
Comparison of Node Centrality Across Career Sages. Note. JT = Joy of Teaching; PV = Professional value; RB = Professional behavior; RV = Role value; SC =School connectedness; NB = Natural acting behavior; RE = Reappraisal; TE = Teaching efficacy; PB = Professional belonging; DB = Deep acting behavior; SU = Suppression; SB = Surface acting behavior.

**Table 1 behavsci-14-01132-t001:** Participants’ Demographic Information.

Teacher-Level			
Variable	Category	N	Percentage
Teaching Grade	K1	691	33.4%
	K2	674	32.6%
	K3	694	33.5%
	Mixed-Age	10	0.5%
Gender	Male	34	1.6%
	Female	2035	98.4%
Position	Leader Teacher	982	47.5%
	Associate Teacher	1087	52.5%
Education Attainment	Secondary School	17	0.8%
	College	509	25.4%
	Undergraduate	1525	73.7%
	Postgraduate	18	0.9%
Nature	Public	83	67.48%
	Private	40	32.52%
Career Stage	The Novice Stage	612	29.58%
	The Advanced Beginners Stage	537	25.95%
	The Competent Stage	920	44.47%
	Age	28.64	5.984
	Working age	6.998	5.256

**Table 2 behavsci-14-01132-t002:** Descriptive Statistics of Study Variables (N = 2069).

	M	SD	Skewness	Kurtosis	1	2	3	4	5	6	7	8	9	10	11
1. SC	3.41	0.53	−0.68	0.32	1										
2. JT	3.50	0.48	−0.79	1.04	0.67 **	1									
3. TE	3.27	0.52	−0.50	0.41	0.45 **	0.54 **	1								
4. SB	2.18	0.88	0.44	−0.65	−0.25 **	−0.30 **	−0.28 **	1							
5. NB	3.15	0.93	−0.51	−0.28	−0.01	−0.03	−0.06 **	0.40 **	1						
6. DB	3.84	0.86	−0.83	0.70	0.22 **	0.23 **	0.19 **	−0.29 **	0.14 **	1					
7. RE	3.99	0.58	−0.57	1.03	0.31 **	0.33 **	0.27 **	−0.16 **	0.12 **	0.29 **	1				
8. SU	2.58	0.88	.28	−0.32	−0.09 **	−0.06 **	−0.01	0.07 **	0.04	−0.01	0.03	1			
9. RV	3.97	0.67	−0.61	0.57	0.46 **	0.52 **	0.47 **	−0.33 **	−0.06 **	0.25 **	0.34 **	0	1		
10. RB	4.27	0.49	−0.55	0.32	0.31 **	0.34 **	0.34 **	−0.18 **	0.02	0.18 **	0.29 **	−0.02	0.56 **	1	
11. PV	4.43	0.55	−0.99	0.85	0.32 **	0.33 **	0.26 **	−0.18 **	0.01	0.19 **	0.28 **	−0.07 **	0.56 **	0.62 **	1
12. PB	3.93	0.75	−0.80	0.79	0.12 **	0.11 **	0.15 **	−0.00	0.06 **	0.06 **	0.11 **	0.03	0.25 **	0.37 **	0.39 **

Note. ** *p* < 0.01. SC = School connectedness; JT = Joy of teaching; TE = Teaching efficacy; SB = Surface acting behavior; NB = Natural acting behavior; DB = Deep acting behavior; RE = Reappraisal; SU = Suppression; RV = Role value; RB = Professional behavior; PV = Professional value; PB = Professional belonging.

**Table 3 behavsci-14-01132-t003:** Comparisons of Emotional Intelligence Factors.

Variable	C1 Novice Stage	C2 Advanced Beginners Stage	C3 Competent Stage	
N = 2069	N = 612	N = 537	N = 920	
	M (SD)	M(SD)	M (SD)	Sig.
SC	3.36 (0.55)	3.40 (0.52)	3.45 (0.51)	3 > 1
JT	3.44 (0.50)	3.49 (0.47)	3.54 (0.47)	3 > 1
TE	3.13 (0.59)	3.29 (0.48)	3.36 (0.48)	3 > 2 > 1
SB	2.24 (0.85)	2.14 (0.87)	2.17 (0.92)	1 = 2 = 3
NB	3.24 (0.86)	3.12 (0.90)	3.12 (0.97)	1 > 3
DB	3.75 (0.84)	3.85 (0.84)	3.89 (0.88)	3 > 1
RE	3.94 (0.60)	3.99 (0.57)	4.04 (0.59)	3 > 1
SU	2.55 (0.84)	2.58 (0.94)	2.60 (0.88)	n.s.
RV	3.81 (0.70)	3.97 (0.65)	4.08 (0.64)	3 > 2 > 1
RB	4.18 (0.51)	4.28 (0.51)	4.33 (0.48)	2 > 1; 3 > 1
PV	4.40 (0.57)	4.41 (0.57)	4.46 (0.54)	n.s.
PB	3.87 (0.79)	3.91 (0.74)	3.99 (0.75)	3 > 2; 3 > 1

Note. Mean differences are significant at *p* < 0.001; equal signs denote insignificance. SC = School connectedness; JT = Joy of teaching; TE = Teaching efficacy; SB = Surface acting behavior; NB = Natural acting behavior; DB = Deep acting behavior; RE = Reappraisal; SU = Suppression; RV = Role value; RB = Professional behavior; PV = Professional value; PB = Professional belonging.

**Table 4 behavsci-14-01132-t004:** Centrality Measures for The Network Based on Whole Sample and Three Stages.

	Strength				Expected Influence				Predictability			
	Whole	C1	C2	C3	Whole	C1	C2	C3	Whole	C1	C2	C3
SC	0.84	0.89	0.70	0.91	0.72	0.87	0.68	0.64	0.22	0.53	0.40	0.40
JT	1.08	1.11	1.14	1.02	0.94	0.91	0.91	0.94	0.03	0.58	0.54	0.54
TE	0.66	0.61	0.65	0.73	0.50	0.50	0.53	0.53	0.54	0.36	0.32	0.32
SB	1.04	0.96	0.97	1.08	−0.11	−0.20	−0.07	−0.07	0.47	0.34	0.31	0.31
NB	0.75	0.66	0.73	0.80	0.75	0.63	0.73	0.80	0.49	0.20	0.26	0.26
DB	0.75	0.76	0.68	0.72	0.20	0.17	0.27	0.18	0.18	0.26	0.19	0.19
RE	0.67	0.54	0.64	0.78	0.64	0.54	0.51	0.78	0.00	0.18	0.18	0.18
SU	0.14	0.09	0.13	0.27	−0.00	−0.03	−0.01	0.08	0.00	0.01	0.01	0.01
RV	1.12	1.13	0.97	1.16	0.86	0.83	0.74	0.93	0.58	0.56	0.44	0.44
RB	0.91	0.89	0.91	0.90	0.91	0.89	0.91	0.90	0.20	0.45	0.48	0.48
PV	0.97	0.96	0.98	0.96	0.93	0.85	0.91	0.93	0.61	0.50	0.47	0.47
PB	0.44	0.49	0.33	0.45	0.44	0.49	0.33	0.45	0.01	0.22	0.16	0.16

Note. C1 = the novice stage; C2 = the advanced beginners stage; C3 = the competent stage; SC = School connectedness; JT = Joy of teaching; TE = Teaching efficacy; SB = Surface acting behavior; NB = Natural acting behavior; DB = Deep acting behavior; RE = Reappraisal; SU = Suppression; RV = Role value; RB = Professional behavior; PV = Professional value; PB = Professional belonging.

**Table 5 behavsci-14-01132-t005:** Results of Network Comparison Test across Career Stages.

Stage	Strength Value	Bonferroni-Adjusted *p* of Global Strength	Bonferroni-Adjusted *p* of Network Invariance
C1 vs. C2	4.570 vs. 4.435	0.724	0.031
C1 vs. C3	4.570 vs. 4.922	0.298	0.022
C2 vs. C3	4.435 vs. 4.922	0.214	0.601

Note. C1 = the novice stage; C2 = the advanced beginners stage; C3 = the competent stage.

**Table 6 behavsci-14-01132-t006:** Results of Significant Test of Centrality Indices in the Network Comparison Test across Career Stages.

Dimension	Indicator	C1 vs. C2		C1 vs. C3		C2 vs. C3	
		XI	Strength	XI	Strength	XI	Strength
Well-Being	SC	1	1	0.623	1	1	1
	JT	1	1	1	1	1	1
	TE	1	1	1	1	1	1
Emotional Labor	SB	1	1	1	1	1	1
	NB	1	1	1	1	1	1
	DB	1	1	1	1	1	1
Emotion Regulation	RE	1	1	0.527	0.599	0.671	1
	SU	1	1	1	1	1	1
Professional Identity	RV	1	1	1	1	1	0.839
	RB	1	1	1	1	1	1
	PV	1	1	1	1	1	1
	PB	1	1	1	1	1	1

Note. XI = Expected influence; C1 = the novice stage; C2 = the advanced beginners stage; C3 = the competent stage; SC = School connectedness; JT = Joy of teaching; TE = Teaching efficacy; SB = Surface acting behavior; NB = Natural acting behavior; DB = Deep acting behavior; RE = Reappraisal; SU = Suppression; RV = Role value; RB = Professional behavior; PV = Professional value; PB = Professional belonging.

## Data Availability

Data are available upon request.

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
