# Peer review of "A Network Analysis of Emotional Intelligence in Chinese Preschool Teachers"

_behavsci, 2024, doi:10.3390/bs14121132_

Round 1
Reviewer 1 Report
Comments and Suggestions for Authors
Reviewer comments:
Manuscript summary Completed
The authors utilized network analysis to examine the relationship between four aspects of emotional intelligence (emotional labor, emotional regulation, professional well-being, and professional identity) in preschool teachers. They asserted the importance of preschool teachers fostering a positive social-emotional classroom environment, as the emotional regulation skills of early childhood educators have a significant impact on young children.
Title and Abstract – overall evaluation
Outstanding.
The abstract captures the readers' attention.
Introduction
Minor modification.
In line 40, the authors indicated that early childhood education spans the age of 3 – 6. However, early childhood education spans the age of birth to age 8.
The authors need to indicate that while early childhood spans the age of birth to 8 years of age, children ages 3 – 6 which is the preschool population were the focus of their study.
Additionally, at the beginning of the introduction, the authors indicated that emotional intelligence will be noted as (EI). In line 49 the authors used EI; however, in lines 37, 46, 51, 53, 55, 59, 60, 64, 66, 67, etc. the authors switched and diverted back to not using (EI).
This needs to be modified because it is important to be consistent with the acronym/abbreviation.
Literature review
The literature review is clearly presented, relevant to the field, and well-structured. The authors included relevant research studies that not only contextualized emotional intelligence but also conceptualized its four aspects.
Methodology / Materials and Methods – overall evaluation
Scientifically sound.
The network analysis design is suitable for assessing teachers' emotional intelligence and addressing the two research questions.
Results/Data Analysis– overall evaluation
Sound.
The manuscript’s data analysis is clearly explained and well-structured. The results section displayed descriptive statistics of variables to summarize the data sets representative of the population as well as, but not limited to correlational networks showing that the strongest connection was between school connectedness and the joy of teaching.
Figures/Tables – overall evaluation
Sound.
The tables and figures in the manuscript are well-suited for presenting the data and are easy to interpret. The authors consistently and accurately interpreted the data while aligning and assessing teachers' emotional intelligence and addressing the two research questions.
Interpretation / Discussion – overall evaluation
Sound.
The authors restated the purpose of the study and emphasized the suitability of network analysis. An important point of discussion highlighted by the authors was the connection between social value, personal growth, and increased satisfaction and pride. Additionally, throughout the discussion section, the authors connected the discussion to the study's findings and the relationship between the four aspects of emotional intelligence (emotional labor, emotional regulation, professional well-being, and professional identity) in preschool teachers.
Conclusions – overall evaluation
Sound.
Statements in the conclusion are coherent and include contributions to the field.
Compliance with Ethical Standards – overall evaluation
Outstanding.
The authors stated that the study obtained approval from the Human Research Ethics Committee at the university. The authors also stated that teachers were invited to participate in the study and consent was obtained before teachers were asked to complete the questionnaires.
References – overall evaluation
Sound.
Cited references are mostly recent publications (within the last 5 years) and relevant.
Writing – overall evaluation
Outstanding, but minor modifications in the introduction section and regarding clarifying the use of emotional intelligence noted as (EI) throughout the manuscript.
Author Response
Please refer to the attachment, thanks!

Reviewer 2 Report
Comments and Suggestions for Authors
Dear authors,
Thank you for your manuscript.
I am of the opinion that this topic is a valuable area of study that addresses a significant research gap and utilizes advanced statistical analysis techniques. However, I would like to address the following issue:
1. Perhaps the authors should provide a justification for the gender imbalance of samples in this study. An explanation would enhance the manuscript.
2. I have some significant concerns about the measurement. The Professional Well-Being construct is using 4 scales instead of the more commonly used 5, 7, or 9 scales, which could compromise the psychometric properties of the instrument. I would like to suggest that the authors justify this by citing literature that uses the same scale, or perhaps by elaborating on the measure based on its pioneering perspective. Additionally, other constructs were using 5 scales, which further contributed to confusion.
Thank you.
Author Response
Please refer to the attachment, thanks!

Reviewer 3 Report
Comments and Suggestions for Authors
Please see the attached file to revise your manuscript.

Author Response
Please refer to the attachment, thank you!

Round 2
Reviewer 3 Report
Comments and Suggestions for Authors
Please see the attached file

It requries slight corrections.
